# Robust compression and detection of epileptiform patterns in ECoG using a real-time spiking neural network hardware framework

Filippo Costa [1,2] ✉, Eline V. Schaft [3], Geertjan Huiskamp[3], Erik J. Aarnoutse[3], Maryse A. van't Klooster [3], Niklaus Krayenbühl [4], Georgia Ramantani [4,5], Maeike Zijlmans [3,6], Giacomo Indiveri [2,5] & Johannes Sarnthein [1,5] ✉

Interictal Epileptiform Discharges (IED) and High Frequency Oscillations (HFO) in intraoperative electrocorticography (ECoG) may guide the surgeon by delineating the epileptogenic zone. We designed a modular spiking neural network (SNN) in a mixed-signal neuromorphic device to process the ECoG in real-time. We exploit the variability of the inhomogeneous silicon neurons to achieve efficient sparse and decorrelated temporal signal encoding. We interface the full-custom SNN device to the BCI2000 real-time framework and configure the setup to detect HFO and IED co-occurring with HFO (IED-HFO). We validate the setup on pre-recorded data and obtain HFO rates that are concordant with a previously validated offline algorithm (Spearman's ρ = 0.75, p = 1e-4), achieving the same postsurgical seizure freedom predictions for all patients. In a remote on-line analysis, intraoperative ECoG recorded in Utrecht was compressed and transferred to Zurich for SNN processing and successful IED-HFO detection in real-time. These results further demonstrate how auto-mated remote real-time detection may enable the use of HFO in clinical practice.

In patients with pharmaco-resistant focal epilepsy, resective epilepsy surgery is a key treatment option[1]. During surgery, intraoperative electrocorticography (ECoG) may serve to delineate the epilepto-genic zone[2]. While interictal epileptiform discharges (IED) in the ECoG are traditionally used to guide surgical decisions, high-frequency oscillations (HFO, 80–500 Hz) have been identified as a new and more precise epileptiform pattern in the intraoperative ECoG; recording HFO may help to tailor epilepsy surgery and improve postoperative seizure outcome[3–5]. Co-occurring IED and

HFO (IED-HFO)[6,7] and Fast Ripple HFO (250–500 Hz)[8] have been shown to indicate epileptogenic tissue with high specificity. The current state-of-the-art approach for intraoperative HFO analysis involves visual annotation by experts either during or after surgery[4] or offline application of automated software algorithms[7,9–12]. Intra-operative HFO analysis that may improve surgical decisions is cur-rently confined to a few epilepsy surgery centers[13]. To make HFO analysis accessible in clinical practice, a standardized analysis that yields recommendations to the surgeon within the duration of the

[1]Klinik für Neurochirurgie, Universitätsspital Zürich und Universität Zürich, Zürich, Switzerland. [2]Institute of Neuroinformatics, University of Zurich and ETH Zurich, Zurich, Switzerland. [3]Department of Neurology and Neurosurgery, University Medical Center Utrecht Brain Center, University Medical Center Utrecht, Utrecht, The Netherlands. [4]Division of Pediatric Neurosurgery, University Children's Hospital Zurich and University of Zurich, Zurich, Switzerland. [5]Zentrum für Neurowissenschaften (ZNZ) Neuroscience Center Zurich, Universität Zürich und ETH Zürich, Zurich, Switzerland. [6]Stichting Epilepsie Instellingen Nederland (SEIN), Heemstede, The Netherlands. ✉e-mail: filippo.costa@usz.ch; johannes.sarnthein@usz.ch

**Fig. 1 | HFO and IED-HFO in ECOG.** We analyzed intraoperative ECoG recorded during resective epilepsy surgery. An HFO is detected in the 250–500 Hz band (red). We process the corresponding activity in the 4–80 Hz EEG band (black) to detect an IED. We define the epileptiform pattern in which an IED co-occurs with an HFO as an IED-HFO. HFO high-frequency oscillation, IED interictal epileptiform discharge.

surgery is needed. Automated real-time detection would significantly reduce human workload and bias.

Due to their event-based processing, spiking neural networks (SNN) are particularly well-suited for bio-signal analysis. Recent work has been done in the detection of HFO with SNN in intracranial EEG[14], ECoG[15,16], and scalp EEG[17]. In particular, HFO detection in the pre-surgical intracranial EEG[14] has been performed using ultra-low-power mixed-signal neuromorphic hardware (DYNAP-SE)[18]. This chip allows the processing of the incoming EEG signal in real time, but it has never been applied in the intraoperative setting. Moreover, the implemented SNN was performing single-channel analysis and did not account for possible artifacts in the EEG that can produce high-amplitude activity in the HFO band. The SNN tested on ECoG data[15,16] introduced an artifact rejection stage, but the network was only simulated in software and therefore it was not suitable for real-time applications.

Novel algorithms have been introduced to train SNN for robust signal processing. Recent works train SNN to minimize the information bottleneck[19–21]. This requires compressing the input with an SNN encoding that preserves task-relevant information. Using the information bottleneck as the main objective, these algorithms train SNN using standard backpropagation through time (BPTT) to train weights. For the DYNAP-SE neuromorphic chip, the use of derivative-free optimization allows training biases that regulate the neural dynamics without explicit definition of the neural model.

Here, we present a real-time setup that comprises fast lightweight algorithms in the open-source and real-time software framework BCI2000[22] and a SNN implemented in DYNAP-SE to detect HFO and IED-HFO patterns. We first validate our algorithm in pre-recorded intraoperative ECoG from 22 patients. HFO rates obtained with our setup are concordant with a previously validated offline algorithm[9] and achieve the same postsurgical seizure freedom predictions for all patients. We then perform a remote real-time analysis with our setup in the intraoperative ECoG of one patient with long-distance data transfer from the University Medical Center Utrecht (UMCU) to the University Hospital Zurich (USZ) where we detect IED-HFO events that are confirmed by expert visual annotations.

The core of the system is the SNN designed in the neuromorphic hardware and inspired by the key concepts of modularity, high-dimensional projection of the signal, and sparse temporal coding. We give a detailed description of the SNN design, showcasing its scalable and parallel multi-channel analysis, the flexible evolutionary algorithm used to optimize the network, and a reconstruction method used to validate the signal compression capabilities of the SNN.

## Results
We first describe the setup that was developed for on-line data analysis in real-time. We used DYNAP-SE to process intraoperative ECoG data in the 4–80 Hz and 250–500 Hz frequency bands to detect HFO and IED-HFO (Fig. 1). In the following we describe each step of the processing pipeline (Fig. 2) in detail.

### Signal preprocessing
To validate a real-time scenario when using pre-recorded data, we used the BCI2000 FilePlayback module at real speed. To stream intraoperative data in real-time, we used the MicromedADC module.

We streamed up to 32 channels in parallel. 32 samples of data (64 ms) were buffered and then processed. Filtering was performed in the traditional EEG frequency band (4–80 Hz) and HFO frequency band (250–500 Hz) with a 64th-order FIR filter. This choice follows the guideline of employing a high filter order for HFO detection[23].

### Delta-modulation encoding
ECoG data were streamed and converted into discrete digital pulses through an Asynchronous Delta Modulator encoding (ADM, Fig. 3) that we implemented in BCI2000 (ADMFilter module). The ADM processing transformed the signal into UP/DN pulses, focusing on epileptiform patterns as events of interest (EoI). Two parameters govern how the ADM operates: the threshold level $\delta$, and the refractory period $\tau$. At the start of the encoding, the first ECoG sample $x(0)$ is set as the baseline, and two thresholds are created. A UP threshold at $x(0) + \delta$, and a DN threshold at $x(0) - \delta$. If the signal crosses one threshold at time $t$, a corresponding UP or DN pulse is produced, and new thresholds are set at $x(t) + \delta$ and $x(t) - \delta$. The minimum allowed inter-pulse interval is equal to the refractory period $\tau$.

The ADMFilter module consisted of an initial tuning phase and an encoding phase. In the tuning phase, streaming data were segmented into non-overlapping windows. The amplitude range of each window was stored, and the ADM threshold level $\delta$ was computed as a pre-defined percentile of the amplitude range distribution. Duration of the tuning phase was set at 5 s, the length of non-overlapping windows was set at 50 ms for the EEG band and 5 ms for the HFO band, the percentile level was set at 40% for the EEG band and at 50% for the HFO band. Parameters were kept fixed for all patients.

In our analysis, the selected threshold level preserved the morphology of IED and HFO, while the signal with amplitude below the threshold was discarded. The ADM processing thus compresses the continuous ECoG trace and is well-suited for remote analyses that require data transfer.

### SNN processing
The discrete UP/DN pulses were then processed with a hardware spiking neural network (SNN) implemented on the Dynamic Asynchronous Neuromorphic Processor (DYNAP-SE)[18] that performed signal compression and focused on EoI (Fig. 4). The silicon neurons in the DYNAP-SE are grouped in four cores of 256 neurons each. Due to their analog nature, the neuron circuits exhibit a variability induced by circuit device mismatch factors that arise during circuit fabrication[24]. Although circuit parameters are shared between all neurons in the same DYNAP-SE core, the device mismatch induced variability produces a distribution of parameters, with shared mean values, but with a coefficient of variation that can be as large as 20%[25]. Although device mismatch is typically perceived as a limitation in analog computation, in our analysis the inherent neural variability is beneficial since it allows processing the incoming ADM pulses with an ensemble of hetero-geneous neurons, which has been shown to improve the information encoding and classification accuracy[26,27].

UP/DN pulses from the EEG and HFO bands were processed separately on two different DYNAP-SE cores (EEG core and HFO core). One ECoG channel was analyzed by four populations of silicon neurons, two in the EEG core and two in the HFO core. Each population was composed of 10 neurons and accumulated activity only from the UP or DN stream. In the following, we refer to these populations as ACC UP and ACC DN. Since high-dimensional projection is useful for pattern separation in the brain, we implemented these two populations in the DYNAP-SE chip to detect epileptiform patterns and reject artifacts. 40 neurons are therefore allocated for the analysis of one channel, 20 for

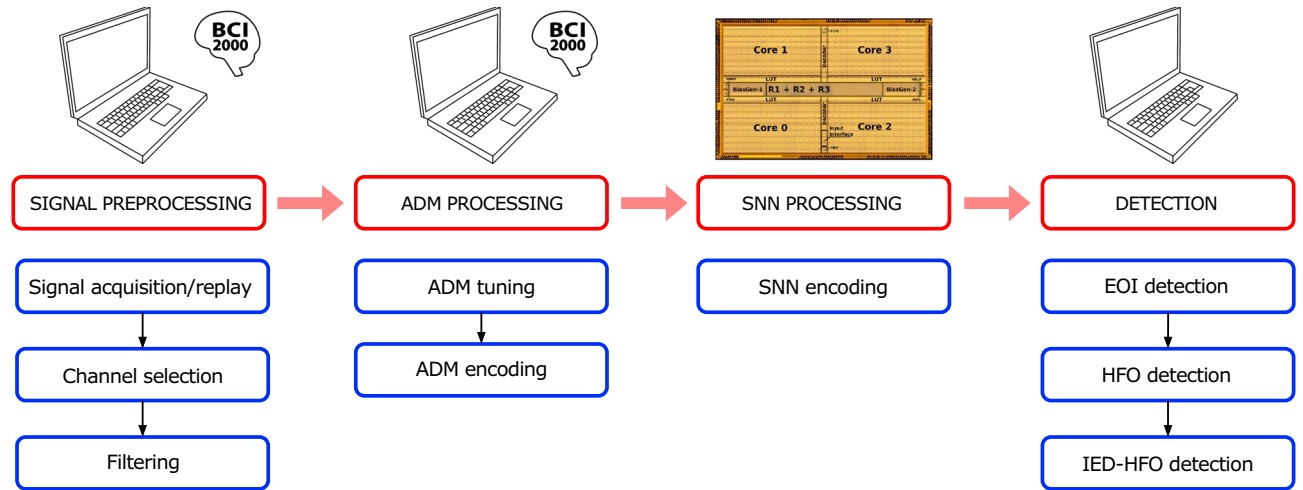

**Fig. 2 | Recording and detection setup.** The detection of epileptiform activity patterns consisted of four steps. 1) The signal preprocessing step was performed in the BCI2000 real-time framework. The ECoG signal was acquired from electrodes in real-time in the intraoperative setting or it was replayed from pre-recorded data to simulate a real-time scenario. A channel selection stage sends the selected channel to further processing steps. A filtering was then performed for each channel in two filter bands, the EEG band (4–80 Hz) and the HFO band (250–500 Hz). The two bands were then sent separately to the ADM processing step. 2) The ADM tuning stage in BCI2000 estimated the baseline amplitude threshold level. The continuous ECoG signal was then encoded into discrete pulses with the ADM encoding. UP/DN pulses were continuously stored in a file that was sent for further processing at the end of the ADM encoding. 3) The stream of pulses was processed by the SNN implemented in the DYNAP-SE chip. 4) The DYNAP-SE output events were sent to an algorithm for the final detection stage. If a pattern of SNN activity fulfilled specific conditions, the underlying signal was denoted as an HFO or an IED-HFO. ADM asynchronous delta modulator, SNN spiking neural network, HFO high-frequency oscillation, EoI event of interest, IED interictal epileptiform discharge.

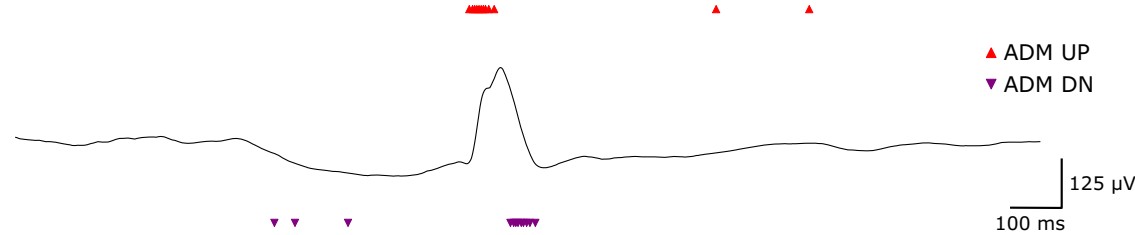

**Fig. 3 | Event-based processing of an IED.** IED and HFO are epileptiform patterns that stand out from the mean noise level. Our ADM encoding processed the signal only when the ECoG amplitude changed by more than a predefined magnitude (delta δ). In this example, the IED generated a dense packet containing a large number of UP/DN pulses. After the IED, the pulse rate went back to the baseline level. ADM event-based processing is an efficient way to compress relevant information in the ECoG trace. ADM asynchronous delta modulator, HFO high-frequency oscillation, IED interictal epileptiform discharge.

the EEG band, and 20 for the HFO band. The DYNAP-SE SNN can process up to 8 channels in parallel, for a total of 160 neurons in the EEG core and 160 neurons in the HFO core. For this use case, we used only two of the four DYNP-SE cores. Moreover, for the analysis of one channel, we are using only 3.9% of the neurons on the chip. The analysis of 8 channels in parallel uses 31.2% of the neurons on the chip.

We used an evolutionary algorithm to train the DYNAP-SE SNN. This derivative-free optimization approach allows training neural hyperparameters as neural and synaptic time constants without an explicit definition of the neural dynamics. This method overcomes the problem of training an SNN with device mismatch, preserving the benefit of using a network with neural heterogeneity. The choice of a population-based optimization allowed us to explore multiple minima regions and to escape local optima.

The evolutionary algorithm selected the best-performing set of parameters, one for HFO and one for IED detection (Fig. 5). The algorithm worked as follows: parameters were sampled from a volume in the parameter space. All the sampled parameter configurations were tested on a 'tuning snippet' that was divided into an IN period, during and shortly after the occurrence of the epileptiform pattern, and two OUT periods, before and long after the occurrence of the epileptiform pattern. A score was associated to each parameter configuration based on the DYNAP-SE activity produced on this snippet. A score was assigned to each neuron $i$ following score $= -\alpha$ |spikes$_{IN}$ - 1| + $\beta$ (spikes$_{IN}$ - spikes$_{OUT}$). spikes$_{IN}$ are all the SNN events that appeared inside the IN period of the tuning snippet. spikes$_{OUT}$ are all the SNN events that appeared in the OUT periods. For $\alpha > \beta > 0$, a neuron received an optimal score if it spiked only one time during the IN period. A global score was then assigned to the neural population based on the mean score over all neurons and the percentage of neurons that produced a spike. Best-performing configurations were set as the centers of the new sampling space. The algorithm proceeded iteratively. The optimal score was associated with a sparse temporal coding in which the SNN produced spikes shortly after the epileptiform pattern. This coding can increase energy efficiency and processing speed, and it is explored in the context of learning in artificial spiking neural networks[28]. The evolutionary algorithm was run as a recalibration step at the beginning of every processing session.

## Detection

We designed simple rules to detect epileptiform patterns from SNN activity. ACC UP and ACC DN activities from EEG and HFO bands were convolved with a linear kernel of duration 100 ms. Contributions from all neurons were summed together, forming one EEG-SNN trace and one HFO-SNN trace. Periods of spiking activity were segmented, and for each period in the HFO-SNN trace, we checked for the presence of

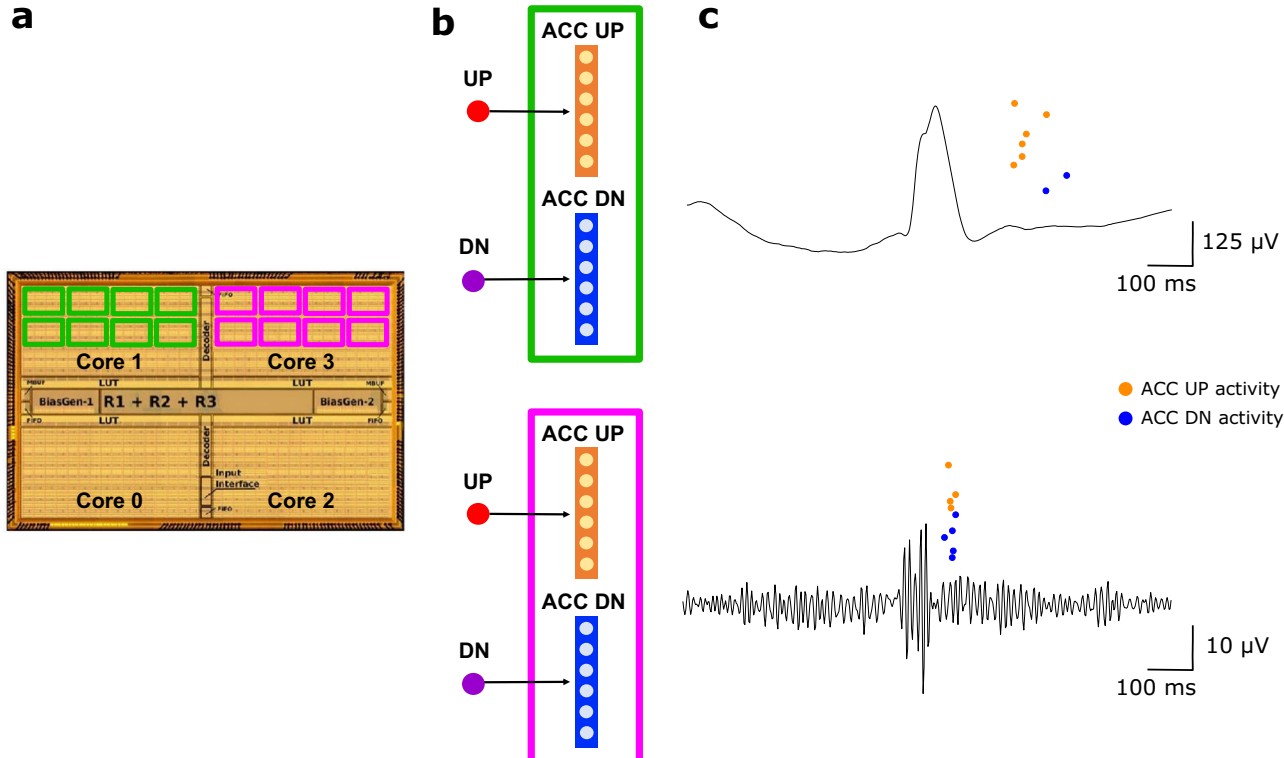

**Fig. 4 | Event-based processing in the neuromorphic chip. a** ADM pulses were processed by two separate cores in the DYNAP-SE neuromorphic chip. One core processed EEG band activity (Core 1), and one processed HFO band activity (Core 3). **b** In each core, a set of identical computational modules processed the ADM signal (green and purple boxes). Each module contained two neural populations: the ACC UP population (orange) received only UP pulses; the ACC DN population (blue) received only DN pulses. The two populations performed a high-dimensional projection of the ADM signal to facilitate artifact rejection. **c** The DYNAP-SE SNN produced output spikes after the occurrence of an IED (top) and an HFO (bottom). ADM asynchronous delta modulator, HFO high-frequency oscillation, IED interictal epileptiform discharge.

both ACC UP and ACC DN activities. If both were present, we defined this period as an EoI-HFO. We classify an EoI-HFO that fulfills these criteria as HFO:

- SNN spikes from the HFO band need to present ≥ 2 neural activations from the ACC UP population, ≥ 2 neural activations from the ACC DN population, and a total contribution of ≥ 6 neurons. This ensures the rejection of low voltage fluctuations.
- Sharp transients in the ECoG produce filtering artifacts. These events produce a spiking pattern where ACC UP and ACC DN activities in the HFO band are separated in time. Therefore, the DYNAP-SE activity in the HFO band needs to present ACC UP and ACC DN activities that are temporally mixed.
- The duration of SNN activity in the HFO band needs to be ≤ 30 ms.
- If SNN activity is present in the EEG band during EoI-HFO occurrence, its duration needs ≤ 500 ms. This step rejects long high-amplitude artifacts in the ECoG trace that may induce high-amplitude activity in the HFO band.

If, together with an HFO, we observe SNN activity in the EEG band with well-separated ACC UP and ACC DN activities and duration ≤ 300 ms, we classify this pattern as IED-HFO.

## Pre-recorded ECoG data in Zurich
We analyzed 22 patients who underwent epilepsy surgery in USZ. The resection was guided by intraoperative high-density ECoG (hd-ECoG, 5 mm contact spacing) for 8 patients and standard grid and strip electrodes (10 mm contact spacing) for 14 patients. The USZ ECoG dataset was previously analyzed for HFO detection with two different detectors, the Spectrum detector and the software SNN (SW-SNN). The Spectrum detector was developed on UMCU intraoperative ECoG recordings[11] and then applied to the USZ intraoperative ECoG data[9,10].

The SW-SNN detector was then applied to the same dataset for the patients whose resection was guided by hd-ECoG[15,16].

Pre-recorded data were streamed with the FilePlayback pipeline in BCI2000 at real speed to simulate a real-time scenario. As an example of the results of our processing pipeline, in the pre-resection ECoG of Patient 5 epileptiform patterns detected by the DYNAP-SE SNN had rates ≥ 1 min⁻¹ only in channels that were later resected (Fig. 6). This was similar to the findings obtained with the Spectrum detector.

The Spearman correlation between the maximum HFO rate for each patient in DYNAP-SE SNN and Spectrum detectors amounted to $\rho = 0.75$ ($p = 1e$-$4$). For all patients, our HFO analysis obtained the same seizure outcome predictions as Spectrum and SW-SNN detectors (Table 1). As for the Spectrum and SW-SNN detectors, in the 8 hd-ECoG patients we obtained PPV = 100%, NPV = 100%, sensitivity = 100%, specificity = 100% and accuracy = 100% (CI [63% 100%]). If we consider all patients, we reached PPV = 100%, NPV = 70%, sensitivity = 25%, specificity = 100%, and accuracy = 73% (CI [50% 89%]). The decrease in sensitivity might arise from the lower spatial resolution of the standard ECoG contact spacing compared to the hd-ECoG. The Spearman correlation between the maximum IED-HFO rate for each patient in the DYNAP-SE SNN and the HFO rate of the Spectrum detector amounted to $\rho = 0.77$ ($p = 7e$-$5$). If we consider all patients, in the IED-HFO analysis we reached PPV = 100%, NPV = 70%, sensitivity = 12.5%, specificity = 100%, and accuracy = 68% (CI [45% 86%]). The additional decrease in sensitivity arises from the lack of IED in the post-resection recording of Patient 6, therefore classified as a FN.

## Real-time analysis of UMCU ECoG in Zurich
To test whether we can use the procedure in real-time, we performed a remote online analysis in collaboration with the UMCU. The ECoG was

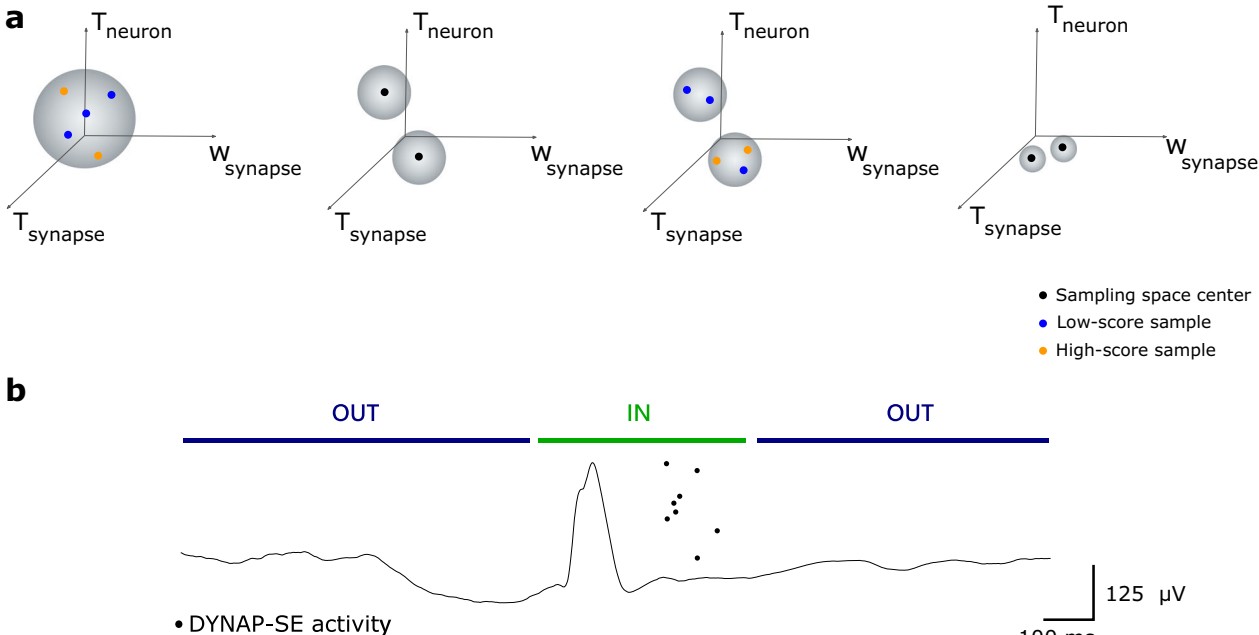

- Sampling space center
- Low-score sample
- High-score sample

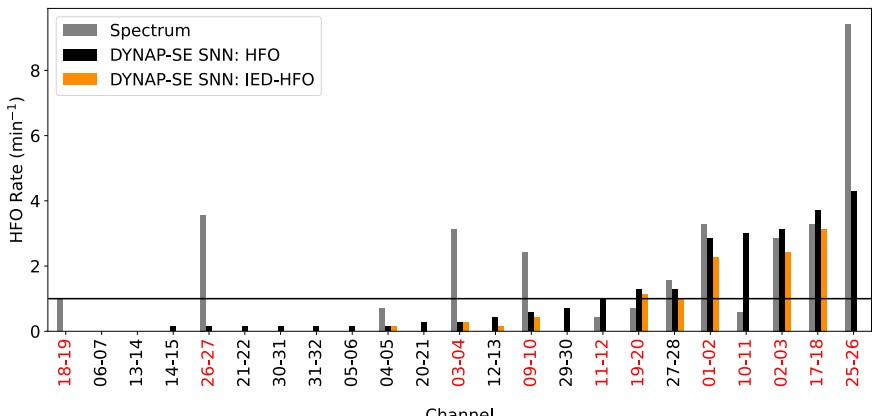

**Fig. 5 | Evolutionary algorithm for DYNAP-SE optimization.** The neuron biases in the DYNAP-SE neuromorphic chip were optimized to produce a sparse temporal encoding of IED and HFO patterns. **a** Parameter configurations were sampled from a volume in the parameter space. Sampled configurations were tested on a tuning snippet. A score was computed for each configuration sample. The configurations with the highest score were set as the centers of the new sampling volumes. The algorithm proceeded iteratively. **b** The tuning snippet was divided into an IN period, during and shortly after the occurrence of the epileptiform pattern, and two OUT periods, before and long after the occurrence of the epileptiform pattern. An optimal score was obtained when SNN neurons spiked only one time during the IN period. HFO high-frequency oscillation, IED interictal epileptiform discharge, SNN spiking neural network.

**Fig. 6 | Comparison of DYNAP-SE SNN and Spectrum detector.** Comparison of DYNAP-SE SNN and Spectrum[9] detectors in Patient 5. The rates obtained with the DYNAP-SE SNN for HFO (black) and IED-HFO (orange) were compared with the HFO rates of the Spectrum detector (gray bars). Most of the channels with high rates were the same for both detectors. In this patient, all the channels with high HFO and IED-HFO rates were recorded from tissue that was later resected (red labels). The patient achieved seizure freedom after surgery and was classified as a true negative (TN). This supports our hypothesis that the detected epileptiform patterns indicate the epileptogenic zone. HFO high-frequency oscillation, IED interictal epileptiform discharge.

recorded in the surgical theater with a Micromed console that was running the Micromed SystemPlus software and streamed the digitized data (2048 Hz) via the UMCU network to a UMCU hospital computer. This computer ran the MicromedADC and the signal processing module in BCI2000 as the real-time framework. There, the ADMFilter performed ADM encoding of the signal into a stream of UP and DN pulses. These lightweight data were then sent to USZ in Zurich, where the DYNAP-SE chip performed further processing (Fig. 7). Two IED-HFO events were detected.

Following the standard UMCU procedure, an expert observer (E.S.) annotated pathological HFO through visual inspection[4]. Both IED-HFO patterns were marked as HFO in the Fast Ripple band (250–500 Hz).

## ECoG compression and reconstruction

The core of our processing strategy involves two compressive steps. First, the continuous ECoG signal $X$ is converted into an ADM encoding $A$ that is then processed by the DYNAP-SE neuromorphic chip to produce the sparse temporal encoding $S$. $X \in \mathbb{R}^T$, where $T$ is the number of sampling points. The ADM encoding is $A \in \{\mathbb{R}^{N_{UP}}, \mathbb{R}^{N_{DN}}\}$, while the SNN encoding is $S \in \{\mathbb{R}^{N_i}\}_{i=1\ldots P}$, where $N_{\mathrm{UP}}$ is the number of UP pulses, $N_{\mathrm{DN}}$ is the number of DN pulses, $N_i$ is the number of spikes from neuron $i$, and P is the number of neurons. The ADM tuning phase sets an amplitude threshold at a predefined percentile of the amplitude range distribution. The SNN training produces a sparse encoding of the ADM input. Therefore, we will always obtain a compression with $T > N_{UP} + N_{DN} > \sum_{i=1}^{P} N_i$. We then tested if the encoding preserves

**Table 1 | Patients**

| Patient | Etiology | Follow up (mo) | Seizure Outcome (ILAE) | Spectrum HFO Rate [min⁻¹][19] | | SW-SNN HFO Rate [min⁻¹][15] | | DYNAP-SE SNN HFO Rate [min⁻¹] | | Spectrum Outcome Prediction | SW-SNN Outcome Prediction | DYNAP-SE SNN Outcome prediction |
|---|---|---|---|---|---|---|---|---|---|---|---|---|
| | | | | Pre | Post | Pre | Post | Pre | Post | | | |
| USZ 1 | DNET | 33 | 1 | 6 | <1 | 3 | <1 | 6 | <1 | TN | TN | TN |
| USZ 2 | FCD 2b | 24 | 1 | 4 | <1 | 10 | <1 | 5 | <1 | TN | TN | TN |
| USZ 3 | Sturge Weber | 30 | 1 | 2 | <1 | 1 | <1 | 1 | <1 | TN | TN | TN |
| USZ 4 | Ganglioglioma | 18 | 1 | 8 | <1 | 12 | <1 | 2 | <1 | TN | TN | TN |
| USZ 5 | FCD 2a | 13 | 1 | 13 | <1 | 30 | <1 | 4 | <1 | TN | TN | TN |
| USZ 6 | Sturge Weber | 20 | 3 | 32 | 5 | 45 | 14 | 10 | 1 | TP | TP | TP |
| USZ 7 | Astrocytoma | 29 | 1 | 1 | <1 | 1 | <1 | 2 | <1 | TN | TN | TN |
| USZ 8 | FCD 2a | 12 | 1 | 22 | <1 | 2 | <1 | 30 | <1 | TN | TN | TN |
| USZ 9 | FCD 2 | 43 | 1 | 1.8 | <1 | - | - | <1 | <1 | TN | - | TN |
| USZ 10 | Sturge Weber | 30 | 3 | <1 | <1 | - | - | <1 | <1 | FN | - | FN |
| USZ 11 | Ependymoma | 28 | 1 | <1 | <1 | - | - | 2 | <1 | TN | - | TN |
| USZ 12 | Anaplastic astrocytoma | 14 | 1 | <1 | <1 | - | - | <<1 | <1 | TN | - | TN |
| USZ 13 | Cavernoma | 29 | 5 | 1.2 | <1 | - | - | 5 | <1 | FN | - | FN |
| USZ 14 | FCD 2b | 16 | 1 | 1.8 | <1 | - | - | 5 | <1 | TN | - | TN |
| USZ 15 | FCD 3 | 35 | 5 | n.a | <1 | - | - | <1 | <1 | FN | - | FN |
| USZ 16 | ODG | 31 | 1 | <1 | <1 | - | - | <1 | <1 | TN | - | TN |
| USZ 17 | FCD 2a | 36 | 3 | n.a | <1 | - | - | <1 | <1 | FN | - | FN |
| USZ 18 | FCD 1a | 38 | 5 | 1.2 | 2.7 | - | - | 7 | 6 | TP | - | TP |
| USZ 19 | Hippocampal sclerosis | 40 | 5 | <1 | <1 | - | - | <1 | <1 | FN | - | FN |
| USZ 20 | FCD 1c | 38 | 1 | 7.0 | <1 | - | - | 5 | <1 | TN | - | TN |
| USZ 21 | Cavernoma | 35 | 5 | <1 | <1 | - | - | <1 | <1 | FN | - | FN |
| USZ 22 | Fibrillary astrocytoma | 40 | 1 | <1 | <1 | - | - | <1 | <1 | TN | - | TN |
| UMCU 23 | Encephalocele | 2 | 1 | - | - | - | - | - | - | - | - | - |

DNET dysembryoplastic neuroepithelial tumor, FCD focal cortical dysplasia, ODG oligodendroglioma, TN true negative, TP true positive, FN false negative, HFO high-frequency oscillation, SW SNN Software SNN.

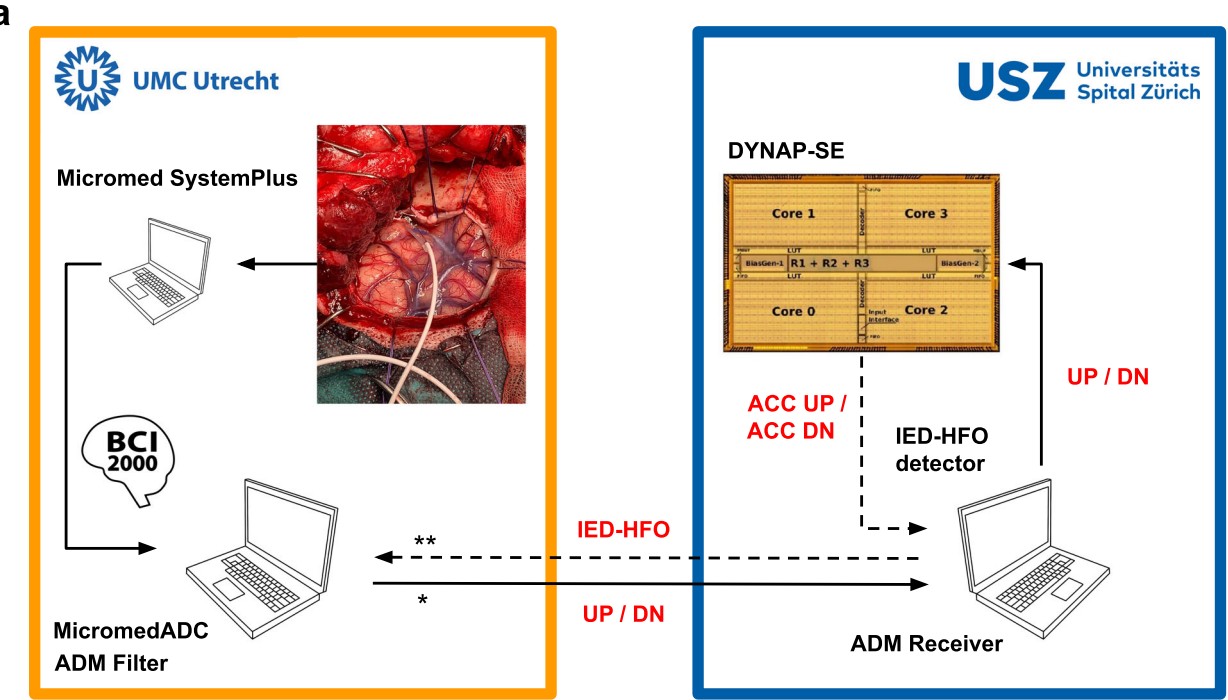

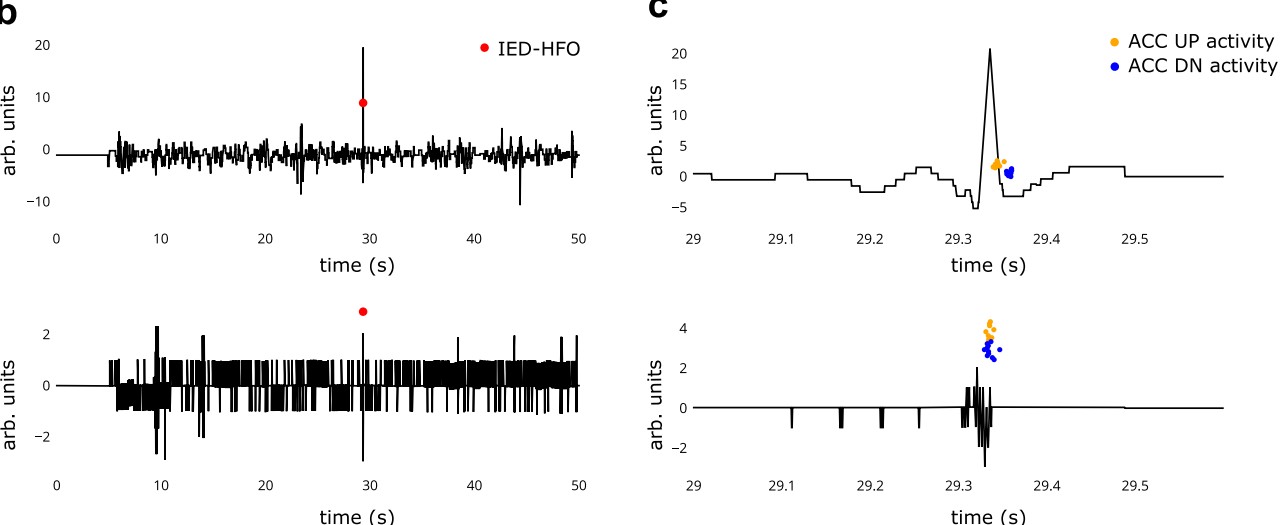

**Fig. 7 | Remote recording setup and IED-HFO detection. a** We performed a remote on-line analysis between UMCU and USZ. Data were recorded during surgery at UMCU with the Micromed SystemPlus software and streamed to a computer running the BCI2000 framework where the ADMFilter module performed the ADM signal encoding. The UP/DN pulses were then sent to USZ for DYNAP-SE processing. Epileptiform patterns were detected based on DYNAP-SE activity. **b** One ECoG channel in the EEG band and in the HFO band reconstructed from UP/DN pulses. We detected as epileptiform patterns only those EoI where HFO and IED co-occurred (red dots). **c** Zoomed view of the epileptiform pattern. DYNAP-SE spikes of the ACC UP (orange dots) and ACC DN (blue dots) populations are produced shortly after the IED and the HFO. This assignment agrees with the expert observer in UMCU (E.S.) who marked this event as a pathological HFO. HFO high-frequency oscillation, IED interictal epileptiform discharge, ADM asynchronous delta modulator.

morphologically relevant information that can be used for signal reconstruction.

Event-based ADM encoding allowed compressing the ECoG trace while preserving the morphology of IED and HFO. As an example we present channel 02-03 in the pre-resection recording of patient 5 with a duration of ~7 min and a sampling rate of 2 kHz. ADM encoding in the EEG band achieved a compression ratio ~20. Given the UP/DN pulses, the ECoG trace $x(t)$ can be reconstructed. We define the reconstruction as $\hat{x}(t)$. Starting from $\hat{x}(0) = x(0)$, if a UP pulse occurs at time $t_{\mathrm{UP}}$,

then $\hat{x}(t_{\mathrm{UP}}) = \hat{x}(t_{\mathrm{UP}} - \varepsilon) + \delta$, where $\varepsilon \to 0$. If a DN pulse occurs at time $t_{\mathrm{DN}}$, then $\hat{x}(t_{\mathrm{DN}}) = \hat{x}(t_{\mathrm{DN}} - \varepsilon) - \delta$.

The hardware SNN performed a high-dimensional projection of the ADM signal and further compressed the ECoG trace. In channel 02-03 of the pre-resection recording of patient 5, the SNN encoding in the EEG band achieved a compression ratio ~34.

To reconstruct the continuous ECoG signal, it is necessary to employ a decoder with memory. RNNs are well-suited for this task and can be trained with standard gradient-based methods such as

**a**

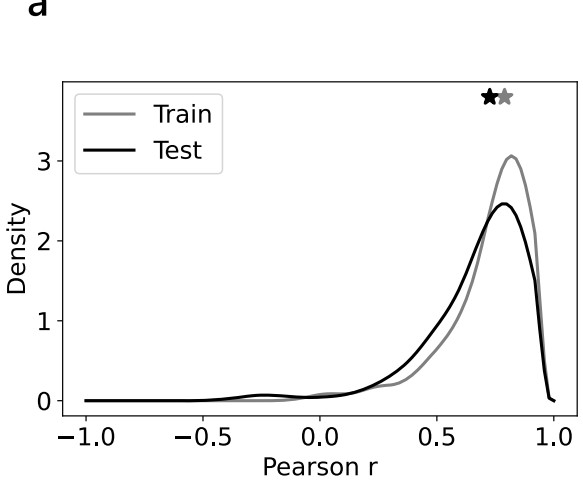

**b**

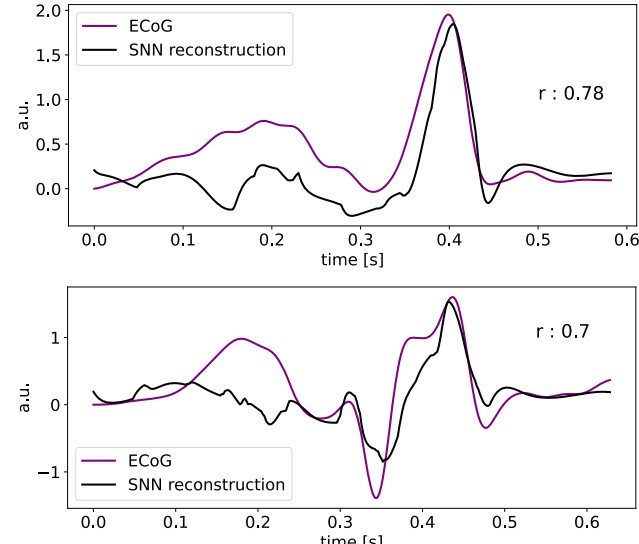

**Fig. 8 | ECoG reconstruction from DYNAP-SE SNN events.** An RNN is trained to reconstruct ECoG snippets from SNN events. **a** Pearson correlation between the ECoG trace and the SNN reconstruction is computed for each snippet in the train and test sets. The correlation distribution for the train set (gray) has a median of 0.79 (gray star). The correlation distribution for the test set (black) has a median of 0.73 (black star). **b** Two reconstruction examples. The SNN reconstruction (black) preserves the main morphological features of the ECoG trace (purple). RNN recurrent neural network, SNN spiking neural network, ECoG electrocorticography.

backpropagation through time (BPTT) or target-based methods. However, especially for long sequences, BPTT training could produce exploding or vanishing gradients. To avoid that, we employed the full-FORCE target-based optimization[29,30]. This method is typically employed to reproduce neural data recordings and is more robust than the standard FORCE learning.

The goal of this reconstruction process is to reproduce the original ECoG trace from SNN activity alone. First, a randomly connected RNN (teacher RNN) received the SNN activity together with the target ECoG trace, while the internal activity of the RNN neurons was recorded. This process ensured that the internal activity combined the information of both input and output. A second network (student RNN) was then trained with a recursive least-squares algorithm to match the internal activity of the teacher RNN when receiving only the SNN activity as input. A linear readout then extracted the target ECoG trace from the student RNN activity.

We trained this reconstruction algorithm on the first 100 seconds of recording from the EEG band of the pre-resection recording of patient 5 in channel 02-03. We then tested on the following 300 seconds. We analyzed the Pearson correlation between the ECoG trace and the SNN reconstruction in snippets where both ACC UP and ACC DN activities were present and the SNN activity was ≤ 500 ms. We obtained similar correlation distributions for our train and test sets, with a median Pearson correlation of 0.79 for the train set and 0.73 for the test set. The reconstruction preserved the main morphological features of the ECoG trace (Fig. 8).

## Discussion

The work presented is at the crossroads of clinical epilepsy research and neuromorphic computing. Leveraging the versatility of the BCI2000 real-time framework, we developed a detection pipeline that makes use of low-power neuromorphic hardware for implementing real-time SNN processing. We applied this approach to HFO and IED-HFO detection in the ECoG trace. We used the BCI2000 real-time framework to stream the ECoG and to realize an ADM encoding that converts the signal into UP/DN pulses, thereby drastically compressing the data. We then fed the UP/DN pulses into the SNN realized in low-power neuromorphic hardware. We finally used the SNN output to detect HFO and IED-HFO. In the validation phase with pre-recorded

data, the results of our setup agreed well with a previously validated offline algorithm. In one UMCU surgery, intraoperative ECoG was compressed in real-time and transferred to Zurich. The detected IED-HFO agreed with the annotation of the human expert. The occurrence of these events can inform the surgeon within the duration of the surgery.

## SNN design principles

We introduce a new DYNAP-SE SNN that succeeds in the real-time intraoperative scenario. The SNN is designed following three main principles (Fig. 4):

- **Modularity.** In our design, we develop a simple computational module composed of two neural populations. Here we are inspired by the modular organization of the cerebral cortex. Replicas of the same module are instantiated across the chip and process signals from different frequency bands and channels. This module is simple since it is based only on feedforward connections. It is interpretable since the SNN output is only related to either UP or DN pulses. It is flexible since it can analyze different frequency bands and channels. It is scalable since it can be replicated across the chip to perform parallel computation.
- **High-dimensional projection.** Here we are inspired by the brain where it is a key principle for pattern separation[31]. In our design, each ACC UP / ACC DN population is composed of 10 neurons and receives input from a single UP or DN pulse stream. While each neuron in one population receives the same input, the inherent analog device mismatch provides the heterogeneity needed to produce decorrelated output. This decorrelation allows setting simple rules for the epileptiform pattern detection based on the temporal mixing of ACC UP and ACC DN activities and the overall duration of the activity in the SNN.
- **Sparse temporal coding.** This principle was exploited in a recent neuromorphic design for odor recognition[32]. In our design, we implement an evolutionary algorithm as a flexible tool to optimize the SNN and to produce a sparse temporal encoding for every epileptiform pattern of interest. Optimization requires only a single snippet of the epileptiform pattern. The score function of the algorithm drives the optimization process, inducing the SNN neurons to spike only once shortly after the occurrence of the

epileptiform pattern. The low output event rate makes this design more efficient than other coding approaches that encode information in the neuron's firing rate. In our application on ECoG, this encoding preserved relevant morphological features of the ECoG signal.

## Main advances over previous approaches

Our study presents an advance over previous neuromorphic computing approaches that have been applied for the detection of HFO in ECoG with a software-simulated SNN[15] and on presurgical intracranial EEG with a hardware SNN[14]. Building on these previous studies, the current study introduces a new scalable computational module that is replicated across the DYNAP-SE chip for epileptiform pattern detection in parallel across 8 channels (Fig. 4). Instead of manually tuning the hardware biases as in the previous approach[14], the SNN is optimized here with an automated flexible evolutionary algorithm that requires only a single tuning snippet for each epileptiform pattern of interest (Fig. 5). This is also an advance over standard machine learning approaches that require large training datasets. Sparse decomposition of ECoG data has been validated to detect pathological HFO[33,34]. We here produce an efficient sparse temporal encoding that allows designing simple rules to detect epileptiform patterns by explicitly using the device mismatch.

Compared to conventional AI chips, in our approach 'the hardware is the software'[35]. The neural heterogeneity, a key feature of the algorithm, is a feature of the chip. Another benefit of using a neuromorphic chip regards data transmission. Contrary to traditional von Neumann architectures, in the DYNAP-SE chip, we only need to transmit the information about firing times. In this application, we only use a shallow network and we can perform pattern detection solely based on the sparse spatio-temporal spiking patterns of this single layer. This highly reduces the power consumption and reduces the data transmission rate, since only spike times and neuron identities are needed to detect an event.

## Measurement perspective

From a measurement perspective, HFO has a rare appearance (a few per minute), short duration (some tens of milliseconds), low amplitude (up to a few µV), and the generators of HFO are confined to a small area of the brain tissue (up to a few mm). All these characteristics make HFO demanding to detect against the noise level. One needs to optimize the signal-to-noise ratio (SNR) for HFO detection in the measurement chain[13]. As one link in the chain, a low noise amplifier has improved HFO detectability[10,36]. As another link in the chain, a higher electrode contact density[9,37] and a lower electrode contact impedance[38] have enhanced HFO detectability. We add here an SNN realized in low-power neuromorphic hardware and embedded in a real-time framework to detect these low-amplitude events.

A research line is adding spatial information to spiking neural networks for biomarker discovery in scalp EEG data recorded with the standardized 10-20 electrode montage[39,40], mapping the signal of each electrode to a neuron in the SNN, preserving spatial relations between nodes and adding plastic recurrent connections to include spatio-temporal interactions. In our work, we optimized the processing for ECoG signals that were acquired with individually placed grid electrodes with an inter-electrode distance of 5 mm. Given the small area of HFO generators[9,37], we performed independent processing for each bipolar channel.

The low sensitivity in predicting poor seizure outcomes must be viewed in the larger perspective of the recording procedure. One aspect is the low signal-to-noise ratio for HFO as a challenge for the recording technology[9,10,13,36-38]. More relevant may be the limited spatial coverage of the recording electrodes. There are good reasons to include only seizure-free patients for the validation of biomarkers[41]: While for seizure-free patients it is clear that the epileptogenic zone

has been identified and removed, for patients with poor outcomes this is less clear. Wrong seizure outcome prediction for patients that are not seizure-free may arise from several reasons, among them the low sensitivity of the biomarker, the misplacement of the electrodes outside the epileptogenic zone, the effects of anesthesia to suppress epileptiform patterns, or the inability of the algorithm to detect the biomarker. All these reasons may contribute to the low sensitivity when predicting poor seizure outcomes.

## Computation at the edge

Compared to conventional AI chips, the neuromorphic approach offers computation with very low power consumption. In particular, as highlighted in a previous publication, the DYNAP-SE neuromorphic chip carries out the HFO processing with a sub mW power budget[14].

In this work, we also show that a key advantage of using neuromorphic chips over low-power AI chips is the event-based processing that leads to low data transmission. To detect the relevant pattern (HFO in the 250–500 Hz band) ECoG has to be recorded with a high sampling rate ($\geq 1$ kHz) leading to large amounts of data which prohibit remote real-time analysis. In the Utrecht-to-Zurich analysis, the event-based ADM method (Fig. 3) performed local computation, resampling the data adaptively and allowing the transfer of lightweight sparse pulses for further processing (Fig. 7).

Our pipeline establishes the computational infrastructure and has the technical expertise available at a single site with other hospitals forwarding their data for processing to this site. Centralizing computing resources at one site ensures that remote hospitals will have access to this analysis. In this work, we establish the pipeline for epileptiform pattern detection 'as a remote service' with low data transmission.

The spike-based temporal encoding in the SNN compresses the signal even further. To validate this second signal compression stage, we applied a signal reconstruction method based on the full-FORCE target-based algorithm[29] that learns to reconstruct the ECoG trace using the SNN events as input and showed that this reconstruction preserves the relevant patterns in the ECoG (Fig. 8).

For the presented remote processing task, and in the outlook of an implantable device that can process the ECoG signal close to the recording site, the neuromorphic approach offers a great advantage in terms of low data rate[42].

While our approach requires training the neuromorphic chip with a computer in the loop, new neuromorphic architectures have been developed and allow on-chip training[21,43-46]. This new generation of neuromorphic chips can bring us closer to implantable brain-machine interfaces with event-based processing and low-power consumption.

We applied the key concepts of modularity, high-dimensional projection, and sparse temporal coding to implement an SNN in the DYNAP-SE hardware. The scalability of the design, the flexibility of the evolutionary algorithm, and the efficiency of the sparse spike-based compression enabled automated intraoperative HFO analysis in real time. With the BCI2000 open-source framework and the lightweight signal transfer, this remote pipeline can easily be implemented in multiple centers, allowing to perform this analysis 'as a service'.

## Methods
### Patients

We included intraoperative subdural ECoG recordings from 23 patients with drug-resistant epilepsy (median age 17 years, range [1–67], 12 females, Table 1). The sex of participants was determined on self-report and sex was not considered in data analysis. 22 patients underwent epilepsy surgery at USZ. The resection was guided by intraoperative high-density ECoG (hd-ECoG) for 8 patients and standard (low-density) grid and strip electrodes for 14 patients, post-resection ECoG was available, and the follow-up duration after surgery was $\geq 12$ months. One patient underwent epilepsy surgery at the

University Medical Center Utrecht (UMCU) where the resection was guided by an intraoperative strip electrode.

## Inclusion and ethics

The collection of patient data and their analysis was approved and performed in accordance with the guidelines and regulations of the local research ethics committees (Kantonale Ethikkommission Zürich 2018-02171, RESPect database, UMCU MREC 18-109 C). All patients and/or their parents provided informed consent to reuse their clinical data for research purposes.

## Anesthesia management

According to the USZ standard protocol for epilepsy surgery, anesthesia was induced with intravenous application of Propofol (1.5–2 mg kg$^{-1}$) and Fentanyl (2–3 µg kg$^{-1}$). Intratracheal intubation was facilitated by Atracurium (0.5 mg kg$^{-1}$), which was stopped afterward to avoid muscle relaxation. Anesthesia was maintained with Propofol (5–10 mg kg$^{-1}$ h$^{-1}$) and Remifentanil (0.1–2 µg kg$^{-1}$ min$^{-1}$). Twenty minutes before ECoG recording, Propofol was ceased and anesthesia was sustained with Sevoflurane (Minimum alveolar concentration (MAC) < 0.5). The depth of anesthesia was monitored by the bispectral index typically in the range 35-40.

According to the UMCU standard protocol for epilepsy surgery, anesthesia was performed with Total Intra-Venous Anesthesia (TIVA). Propofol was used to induce and maintain anesthesia and combined with analgesics (Sulfentanil or Remifentanil) and muscle relaxants (e.g., Rocuronium). Just before ECoG recording, anesthesiologists paused the Propofol administration, while the administration of analgesics was continued. Propofol was restarted after 5–15 min.

## Recording setup

At USZ, hd-ECoG was recorded using high-density subdural grid electrodes in a 4 × 8 electrode array (Ad-Tech Medical, https://adtechmedical.com/). The hd-ECoG electrodes had a contact exposure diameter of 2.3 mm and an inter-electrode distance of 5 mm. The standard ECoG electrodes had a contact exposure diameter of 5 mm and an inter-electrode distance of 10 mm. We used a needle electrode placed in the dura as an electrical reference. We recorded the continuous ECoG for offline processing (Nicolet CSeries amplifier, natus.com, AD conversion 16 bit, sampling rate 2 kHz, 1–800 Hz pass band). All ECoG data was re-referenced to a bipolar montage along the length of the grid.

At UMCU, ECoG was recorded with a subdural strip electrode with 1 × 6 contacts, a contact exposure diameter of 2.3 mm, and an inter-electrode distance of 10 mm (Ad-Tech Medical) with a Micromed amplifier (micromedgroup.com, AD conversion 16 bit, sampling rate 2048 Hz, 0.15 Hz high pass) and Micromed software (SystemPlus).

## Statistics

In the clinical validation of the results, we follow the same approach as refs. 9,15. Following ref. 4, we defined:

- True positive (TP): a patient where the post-resection ECoG contained HFO with a rate ≥ 1 min$^{-1}$ and the patient was not seizure-free after surgery (ILAE 2–6).
- False positive (FP): a patient where the post-resection ECoG contained HFO with a rate ≥ 1 min$^{-1}$ and the patient was seizure-free after surgery (ILAE 1).
- False negative (FN): a patient where the post-resection ECoG contained HFO with a rate < 1 min$^{-1}$ and the patient was not seizure-free after surgery (ILAE 2-6).
- True negative (TN): a patient where the post-resection ECoG contained HFO with a rate < 1 min$^{-1}$ and the patient was seizure-free after surgery (ILAE 1).

The positive predictive value was calculated as PPV= TP/(TP + FP), negative predictive value as NPV = TN/(TN + FN), sensitivity = TP/(TP + FN), specificity = TN/(TN + FP), and accuracy = (TP + TN)/*N*. Statistical analysis was performed with the Scipy Python package. We used the Clopper-Pearson method for estimates of the 95% confidence intervals (CI). We used the two-sided Spearman's method to calculate the correlation. Statistical significance was established at *p* < 0.05.

## Reporting summary

Further information on research design is available in the Nature Portfolio Reporting Summary linked to this article.

## Data availability

The pre-recorded ECoG with BCI2000-compatible data, ADM encoding, SNN encoding, and epileptiform pattern markings, are available in the OpenNeuro database (https://doi.org/10.18112/openneuro.ds004944.v1.1.0).

## Code availability

The custom ADM module, a demo to run ADM conversion, and the code to detect epileptiform patterns from DYNAP-SE activity are provided at https://doi.org/10.5281/zenodo.10959559. (Windows: BCI2000 3.6.7010, Python 3.10.11; Linux: Python 3.8.10).

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

## Acknowledgements

We thank V. Dimakopoulos for help in reformatting the data to BIDS and uploading the data in OpenNeuro. We acknowledge grants awarded by the Swiss National Science Foundation (funded by the SNSF 204651 to J.S. and G.I. with G.R. as project partner, and by SNSF 208184 to G.R. with J.S. as project partner) and by the European Research Council (ERC starting grant 803880 to M.Z. that funded M.Z., E.S., M.v.t.K), the Anna Mueller Grocholski Foundation, and the Vontobel Foundation to G.R. The funders had no role in the design or analysis of the study.

## Author contributions

F.C., J.S., M.Z., E.A., G.H., G.I. designed the experiments. F.C. set up the data analysis pipeline and performed data analysis. G.R., N.K., M.Z. provided patient care. E.S., M.v.t.K were responsible for data collection in UMCU. F.C. did statistics and prepared the figures. F.C., J.S., G.R., M.v.t.K. wrote the manuscript. All authors approved the final version of the manuscript.

## Competing interests

The authors declare no competing interests.
