## [Peer Review File · Nature Communications]

REVIEWER COMMENTS

Reviewer #1 (Remarks to the Author):

The work presented is at the crossroads of clinical epilepsy research and neuromorphic computing. Leveraging the versatility of the BCI2000 real-time framework, we developed a detection pipeline that makes use of a low-power neuromorphic hardware for implementing a real-time SNN processing. We applied this approach to HFO and IED-HFO detection in the ECoG trace. The paper is organized well. However, there are some points to be considered.

1. When introducing SNNs, some novel representative algorithm work should be included. They are presented for robust learning and robust signal processing for neuromorphic hardware. They are also suitable and promising for ECoG compression and detection. They include: SNIB: Improving Spike-Based Machine Learning Using Nonlinear Information Bottleneck; Effective Surrogate Gradient Learning With High-Order Information Bottleneck for Spike-Based Machine Intelligence; SIBoLS: Robust and Energy-efficient Learning for Spike-based Machine Intelligence in Information Bottleneck Framework.
2. Please further discuss the advantages of neuromorphic hardware to deal with the ECoG signals in comparison with the conventional AI chips, especially low-power AI chips.
3. Please also include some novel work in neuromorphic architecture design, including: NADOL: Neuromorphic Architecture for Spike-driven Online Learning By Dendrites; Integrating Visual Perception With Decision Making in Neuromorphic Fault-Tolerant Quadruplet-Spike Learning Framework.
4. What is the reason to select the used evolutionary algorithm other than others?
5. Why is the full-FORCE algorithm is used for the ECoG reconstruction? Is there any other suitable algorithms?
6. What is the hardware resource usage report? How does the power consumption perform?

Reviewer #2 (Remarks to the Author):

The noteworthy results are the use of neuromorphic chip to analyse ECoG signal sent from a remote site and to send response to surgeons on-line.

This is a pioneering work in the field.

The work is novel when compared to established literature.

The work supports the conclusions and claims, but additional evidence is needed.

I do not think there are flaws in the data analysis, interpretation and conclusions.

The methodology is sound, still it can be improved.

The work is original, novel and it is worth publishing.

This work can be significantly improved and it has to be published in the future work section as well:

1) Why ECoG data from patients under operation need to be sent remotely, while a cheap and low energy neuromorphic chip can be placed at any surgery

2) The used neuromorphic chip needs to account for both spatial coordinates and temporal data of the ECoG surgery in a real time. Such methods are proposed in:

1. Diagnostic biomarker discovery from brain EEG data with LSTM, reservoir-SNN and NeuCube: Methods and a pilot study on epilepsy vs migraine, IEEE archive, <https://doi.org/10.36227/techrxiv.23514486.v1>

2. Design of MRI Structured Spiking Neural Networks and Learning Algorithms for Personalized Modelling, Analysis, and Prediction of EEG Signals, Nature, Scientific Reports, June (2021) 11:12064, <https://doi.org/10.1038/s41598-021-90029-5>

3. Time-Space, Spiking Neural Networks and Brain-Inspired Artificial Intelligence, Springer Nature (2019) 750p., <https://www.springer.com/gp/book/9783662577134>

The question is if the proposed neuromorphic system takes into account both the spatial and the temporal information from the ECoG data over time.

This is an important question that needs to be addressed in the paper before publication as an operation happens in both time and space and both are important components in the decision making process in a real time.

REVIEWER COMMENTS

We thank both expert Reviewers for their thoughtful and constructive comments, which have undoubtedly improved the quality of our manuscript. Here, we provide a point-by-point response to all comments. Our responses are in blue, with text that was in the original submission in *blue italics*, and new text in red.

1 Reviewer #1:

The work presented is at the crossroads of clinical epilepsy research and neuromorphic computing. Leveraging the versatility of the BCI2000 real-time framework, we developed a detection pipeline that makes use of a low-power neuromorphic hardware for implementing a real-time SNN processing. We applied this approach to HFO and IED-HFO detection in the ECoG trace. The paper is organized well. However, there are some points to be considered.

1.1 Other algorithms

When introducing SNNs, some novel representative algorithm work should be included. They are presented for robust learning and robust signal processing for neuromorphic hardware. They are also suitable and promising for ECoG compression and detection. They include: SNIB: Improving Spike-Based Machine Learning Using Nonlinear Information Bottleneck; Effective Surrogate Gradient Learning With High-Order Information Bottleneck for Spike-Based Machine Intelligence; SIBoLS: Robust and Energy-efficient Learning for Spike-based Machine Intelligence in Information Bottleneck Framework.

Response: We agree with the Reviewer's comment that the algorithms presented in the referenced publications contribute to providing a better contextual framework for our results. In our revised manuscript, we have added an introduction to the key ideas presented in those publications, focusing on training Spiking Neural Networks (SNN) with information bottleneck.

Action: Following the Reviewer's comment, we now set our algorithm in context with the publications mentioned by the Reviewer.

We have added one paragraph in the Introduction:

Novel algorithms have been introduced to train SNN for robust signal processing. Recent works train SNN minimizing the information bottleneck (Yang and Chen, 2023a, 2023b, Yang, Wang and Chen, 2023). This requires compressing the input with an SNN encoding that preserves task-relevant information. Using the information bottleneck as main objective, these algorithms train SNN using standard backpropagation through time (BPTT) to train weights. For the DYNAP-SE neuromorphic chip, the use of derivative-free optimization allows training biases that regulate the neural dynamics without explicit definition of the neural model.

We have added one paragraph on information compression in the Results 2.8:

The core of our processing strategy involves two compressive steps. First, the continuous ECoG signal X is converted into an ADM encoding A that is then processed by the DYNAP-SE neuromorphic chip to produce the sparse temporal encoding S . $X \in \mathbb{R}^T$, where T is the number of sampling points. The ADM encoding is $A \in \{\mathbb{R}^{N_{UP}}, \mathbb{R}^{N_{DN}}\}$, while the SNN encoding is $S \in \{\mathbb{R}^{N_i}\}_{i=1 \dots P}$, where N_{UP} is the number of UP pulses, N_{DN} is the number of DN pulses, N_i is the number of spikes from neuron i , and P is the number of neurons. The ADM tuning phase sets an amplitude threshold at a predefined percentile of the amplitude range distribution. The SNN training produces a sparse encoding of the ADM input. Therefore, we will always obtain a compression with $T > N_{UP} + N_{DN} > \sum_{i=1}^P N_i$. We then tested if the encoding preserves morphologically relevant information that can be used for signal reconstruction.

1.2 Neuromorphic advantage

Please further discuss the advantages of neuromorphic hardware to deal with the ECoG signals in comparison with the conventional AI chips, especially low-power AI chips.

Response: We thank the Reviewer for the opportunity to discuss in detail the advantages of using a neuromorphic hardware for the analysis of ECoG signals.

Action: We have added a paragraph in Discussion 3.2, presenting a detailed discussion on the advantages of using neuromorphic hardware compared to conventional AI chips for epileptiform pattern detection.

Compared to conventional AI chips, in our approach 'the hardware is the software' (Laydevant, Wright, Wang et al., 2023). The neural heterogeneity, key feature of the algorithm, is a feature of the chip. Another benefit of using a neuromorphic chip regards data transmission. Contrary to traditional von Neumann architectures, in the DYNAP-SE chip we only need to transmit the information about firing times. In this application, we only use a shallow network and we can perform pattern detection solely based on the sparse spatio-temporal spiking patterns of this single layer. This highly reduces the power consumption and reduces the data transmission rate, since only spike times and neuron identities are needed to detect an event.

We rephrased a paragraph in Discussion 3.4 to highlight the advantage of using neuromorphic hardware.

In this work we also show that a key advantage of using neuromorphic chips over low power AI chips is the event-based processing that leads to low data transmission. *To detect the relevant pattern (HFO in the 250-500 Hz band) ECoG has to be recorded with high sampling rate (≥ 1 kHz) leading to large amounts of data which prohibit remote real-time analysis. In the Utrecht-to-Zurich analysis, the event-based ADM method (Figure 3) performed local computation, resampling the data adaptively and allowing the transfer of lightweight sparse pulses for further processing (Figure 7), to achieve a compressed lightweight encoding using UP/DN pulses that were transferred for final processing in our Utrecht-to-Zurich analysis (Figure 7).*

1.3 Neuromorphic hardware

Please also include some novel work in neuromorphic architecture design, including: NADOL: Neuromorphic Architecture for Spike-driven Online Learning By Dendrites; Integrating Visual Perception With Decision Making in Neuromorphic Fault-Tolerant Quadruplet-Spike Learning Framework.

Response: We thank the Reviewer for pointing out novel work in neuromorphic architecture design. We think that including new architectures that can perform on-chip learning can be beneficial for the manuscript, because, in the outlook of an implantable brain-machine interface, new hardware will have to adapt to the patient's needs.

Action: We have added a paragraph in Discussion 3.4 that includes the mentioned publications and new neuromorphic hardware designs with on-chip learning.

While our approach requires training the neuromorphic chip with a computer in the loop, new neuromorphic architectures have been developed and allow on-chip training (Cartiglia, Rubino, Narayanan et al., 2022, Richter, Wu, Whatley et al., 2023, Rubino, Cartiglia, Payvand et al., 2023, Yang, Wang and Chen, 2023, Yang, Wang, Pang et al., 2023). This new generation of neuromorphic chips can bring us closer to implantable brain-machine interfaces with event-based processing and low power consumption.

1.4 Evolutionary algorithm

What is the reason to select the used evolutionary algorithm other than others?

Response: We thank the Reviewer for encouraging us to discuss the reasons to employ a derivative-free optimization method as the presented evolutionary algorithm for training our hardware SNN. We opted for the evolutionary algorithm because training a Spiking Neural Network (SNN) on DYNAP-SE involves optimizing the biases that govern the analog circuit's dynamics. In our specific application, we do not train weights, but we optimize the neuron time constant, the synapse time constant, and the firing threshold. Employing derivative-free optimization enabled us to train the SNN without the need to specify the details of the DYNAP-SE neural dynamics for gradient computation.

Action: We added a paragraph in Introduction:

Novel algorithms have been introduced to train SNN for robust signal processing. Recent works train SNN minimizing the information bottleneck (Yang and Chen, 2023a, 2023b, Yang, Wang and Chen, 2023). This requires compressing the input with an SNN encoding that preserves task-relevant information. Using the information bottleneck as main objective, these algorithms train SNN using standard backpropagation through time (BPTT) to train weights. For the DYNAP-SE neuromorphic chip, the use of derivative-free optimization allows training biases that regulate the neural dynamics without explicit definition of the neural model.

and in Results 2.3:

We used an evolutionary algorithm to train the DYNAP-SE SNN. This derivative-free optimization approach allows training neural hyperparameters as neural and synaptic time constants without explicit definition of the neural dynamics. This method overcomes the problem of training a SNN with device mismatch, preserving the benefit of using a network with neural heterogeneity. The choice of a population-based optimization allowed us to explore multiple minima regions and to escape local optima.

1.5 Full-FORCE

Why is the full-FORCE algorithm used for the ECoG reconstruction? Is there any other suitable algorithms?

Response: The Reviewer points to an interesting decision that we took during our analysis. For ECoG reconstruction, we could train an RNN with a gradient based approach as backpropagation through time (BPTT) or a target-based approach. We opted for the full-FORCE target-based approach to mitigate the risks of exploding or vanishing gradients. We chose the full-FORCE and not the standard FORCE learning for its superior performance and robustness.

Action: We added a paragraph in Results 2.8:

To reconstruct the continuous ECoG signal, it is necessary to employ a decoder with memory. RNN are well suited for this task and can be trained with standard gradient-based methods as backpropagation through time (BPTT) or target-based methods. However, especially for long sequences, BPTT training could produce exploding or vanishing gradients. To avoid that, we employed the full-FORCE target-based optimization (DePasquale, Cueva, Rajan et al., 2018, Liu, Losonczy and Liao, 2022).

This method is typically employed to reproduce neural data recordings and is more robust than the standard FORCE learning.

1.6 Power consumption

What is the hardware resource usage report? How does the power consumption perform?

Response: The Reviewer addresses a point that we want to highlight in our manuscript. Presenting a remote “as a service” use case, power consumption is not a critical factor, while the low data transmission rate is of utmost importance. However, previous work using the same chip showed that the system operates with sub mW budget to detect HFO in the intracranial EEG (Sharifshazileh, Burelo, Sarnthein et al., 2021).

We had described hardware usage in Results 2.3:

One ECoG channel was analyzed by four populations of silicon neurons, two in the EEG core and two in the HFO core. Each population was composed of 10 neurons and accumulated activity only from the UP or DN stream. In the following, we refer to these populations as ACC UP and ACC DN. Since high-dimensional projection is useful for pattern separation in the brain, we implemented these two populations in the DYNAP-SE chip to detect epileptiform patterns and reject artifacts. 40 neurons are therefore allocated for the analysis of one channel, 20 for the EEG band, and 20 for the HFO band. The DYNAP-SE SNN can process up to 8 channels in parallel, for a total of 160 neurons in the EEG core and 160 neurons in the HFO core.

Action: To add information about the power consumption, we added a paragraph in Discussion 3.4:

Compared to conventional AI chips, the neuromorphic approach offers computation with very low power consumption. In particular, as highlighted on a previous publication, the DYNAP-SE neuromorphic chip carries out the HFO processing with a sub mW power budget (Sharifshazileh, Burelo, Sarnthein et al., 2021).

We now provide a more detailed description of the hardware resource usage in Results 2.3:

For this use case, we used only two of the four DYNAP-SE cores. Moreover, for the analysis of one channel we are using only 3.9% of the neurons on the chip. The analysis of 8 channels in parallel uses 31.2% of the neurons on the chip.

2 Reviewer #2:

The noteworthy results are the use of neuromorphic chip to analyse ECoG signal sent from a remote site and to send response to surgeons on-line.

This is a pioneering work in the field.

The work is novel when compared to established literature.

The work supports the conclusions and claims, but additional evidence is needed.
I do not think there are flaws in the data analysis, interpretation and conclusions.
The methodology is sound, still it can be improved.
The work is original, novel and it is worth publishing.
This work can be significantly improved and it has to be published in the future work section as well:

2.1 Remote recording

Why ECoG data from patients under operation need to be sent remotely, while a cheap and low energy neuromorphic chip can be placed at any surgery.

Response: We agree with the Reviewer's comment that it would be effective to use a low-power chip directly in the operating room, for each operation and in each hospital. However, in this work we wanted to validate the "remote as a service" approach for two reasons. First, we use a research platform prototype that uses a complex setup which cannot easily be brought in the operating room. Second, we wanted to verify that this "remote mode" can be a viable option also for hospitals that do not have access to this technology. Even worldwide, the number of surgeries that require intraoperative ECoG is small. Intraoperative ECoG is required only for specific indications in a subset of surgeries performed in epilepsy centers. Even high-load epilepsy centers will not require an ECoG on a daily basis. All these hospitals can forward their data for processing at a remote center that has the computational infrastructure and the technical expertise available. Centralizing resources will ensure that both high-load and low-load hospitals will have (remote) access to qualified personnel and benefit from this advanced analysis pipeline.

Action: Following the Reviewer's comment, we have now added the following paragraph to the Discussion 3.4:

Our pipeline establishes the computational infrastructure and have the technical expertise available at a single site with other hospitals forwarding their data for processing to this site. Centralizing computing resources at one site ensures that remote hospitals will have access to this analysis. In this work we establish the pipeline for epileptiform pattern detection 'as a remote service' with low data transmission.

2.2 EEG references and spatial patterns

The used neuromorphic chip needs to account for both spatial coordinates and temporal data of the ECoG surgery in a real time. Such methods are proposed in:

1. Diagnostic biomarker discovery from brain EEG data with LSTM, reservoir-SNN and NeuCube: Methods and a pilot study on epilepsy vs migraine, IEEE archive, <https://doi.org/10.36227/techrxiv.23514486.v1>
2. Design of MRI Structured Spiking Neural Networks and Learning Algorithms for Personalized Modelling, Analysis, and Prediction of EEG Signals, Nature, Scientific Reports, June (2021) 11:12064, <https://doi.org/10.1038/s41598-021-90029-5>
3. Time-Space, Spiking Neural Networks and Brain-Inspired Artificial Intelligence, Springer Nature (2019) 750p., <https://www.springer.com/gp/book/9783662577134>

The question is if the proposed neuromorphic system takes into account both the spatial and the temporal information from the ECoG data over time.

This is an important question that needs to be addressed in the paper before publication as an operation happens in both time and space and both are important components in the decision making process in a real time.

Response: The Reviewer points us to publications that analyze scalp EEG and capitalize on the standardized locations where EEG electrodes are usually placed. The standardized locations of the 10-20 electrode montage cover the whole brain.

However, when recording Electrocorticography (ECoG) during surgery, the situation is different. The surgical field where the skull is removed usually has a diameter of only a few cm in order to assure safety and sterility of the patient's brain. Our algorithm was optimized for electrode grids that carry 32 recording contacts on the area of 2cm*4cm. Given the contact spacing of only 5 mm and their direct placement on the cortex, the ECoG signal is generated only from brain tissue within mm of the electrode contacts. This close proximity between epilepsy generator and recording contacts is paramount for the surgeon's decision to resect or not

to resect. The role of ECoG is to analyze not standardized spatial patterns but rather the time series of well-defined recording channels to support the surgeon's decision.

Action: Following the Reviewer's comment, we have now added the following paragraph in Discussion 3.3: A research line is adding spatial information to spiking neural networks for biomarker discovery in scalp EEG data recorded with the standardized 10-20 electrode montage (Kasabov, 2023, Kasabov, 2019, Saeedinia, Jahed-Motlagh, Tafakhori et al., 2021), mapping the signal of each electrode to a neuron in the SNN, preserving spatial relations between nodes and adding plastic recurrent connections to include spatio-temporal interactions. In our work, we optimized the processing for ECoG signals that were acquired with individually placed grid electrodes with an inter-electrode distance of 5 mm. Given the small area of HFO generators (Boran, Ramantani, Krayenbuhl et al., 2019, Zweiphenning, van Diessen, Aarnoutse et al., 2020), we performed independent processing for each bipolar channel.

3 References

- Boran E, Ramantani G, Krayenbuhl N, Schreiber M, König K, Fedele T, et al. High-density ECoG improves the detection of high frequency oscillations that predict seizure outcome. *Clinical Neurophysiology* 2019;130(10):1882-8.
- Cartiglia M, Rubino A, Narayanan S, Frenkel C, Haessig G, Indiveri G, et al. Stochastic dendrites enable online learning in mixed-signal neuromorphic processing systems. *2022 IEEE International Symposium on Circuits and Systems (ISCAS)*: IEEE; 2022. p. 476-80.
- DePasquale B, Cueva CJ, Rajan K, Escola GS, Abbott L. full-FORCE: A target-based method for training recurrent networks. *PloS one* 2018;13(2):e0191527.
- Kasabov N. Diagnostic biomarker discovery from brain EEG data with LSTM, reservoir-SNN and NeuCube: Methods and a pilot study on epilepsy vs migraine2023.
- Kasabov NK. *Time-space, spiking neural networks and brain-inspired artificial intelligence*: Springer, 2019.
- Laydevant J, Wright LG, Wang T, McMahon PL. The hardware is the software. *Neuron* 2023.
- Liu LB, Losonczy A, Liao Z. tension: A Python package for FORCE learning. *PLOS Computational Biology* 2022;18(12):e1010722.
- Richter O, Wu C, Whatley AM, Köstinger G, Nielsen C, Qiao N, et al. DYNAP-SE2: a scalable multi-core dynamic neuromorphic asynchronous spiking neural network processor. *arXiv preprint arXiv:231000564* 2023.
- Rubino A, Cartiglia M, Payvand M, Indiveri G. Neuromorphic analog circuits for robust on-chip always-on learning in spiking neural networks. *2023 IEEE 5th International Conference on Artificial Intelligence Circuits and Systems (AICAS)*: IEEE; 2023. p. 1-5.
- Saeedinia SA, Jahed-Motlagh MR, Tafakhori A, Kasabov N. Design of MRI structured spiking neural networks and learning algorithms for personalized modelling, analysis, and prediction of EEG signals. *Scientific Reports* 2021;11(1):12064.
- Sharifshazileh M, Burelo K, Sarnthein J, Indiveri G. An electronic neuromorphic system for real-time detection of high frequency oscillations (HFO) in intracranial EEG. *Nat Commun* 2021;12(1):3095.
- Yang S, Chen B. Effective Surrogate Gradient Learning With High-Order Information Bottleneck for Spike-Based Machine Intelligence. *IEEE Transactions on Neural Networks and Learning Systems* 2023a.
- Yang S, Chen B. SNIB: improving spike-based machine learning using nonlinear information bottleneck. *IEEE Transactions on Systems, Man, and Cybernetics: Systems* 2023b.
- Yang S, Wang H, Chen B. SIBoLS: Robust and Energy-efficient Learning for Spike-based Machine Intelligence in Information Bottleneck Framework. *IEEE Transactions on Cognitive and Developmental Systems* 2023.
- Yang S, Wang H, Pang Y, Jin Y, Linares-Barranco B. Integrating Visual Perception With Decision Making in Neuromorphic Fault-Tolerant Quadruplet-Spike Learning Framework. *IEEE Transactions on Systems, Man, and Cybernetics: Systems* 2023.
- Zendrikov D, Solinas S, Indiveri G. Brain-inspired methods for achieving robust computation in heterogeneous mixed-signal neuromorphic processing systems. *Neuromorphic Computing and Engineering* 2023;3(3):034002.
- Zweiphenning WJEM, van Diessen E, Aarnoutse EJ, Leijten FSS, van Rijen PC, Braun KPJ, et al. The resolution revolution: Comparing spikes and high frequency oscillations in high-density and standard intra-operative electrocorticography of the same patient. *Clinical Neurophysiology* 2020;131(5):1040-3.

REVIEWERS' COMMENTS

Reviewer #1 (Remarks to the Author):

I am satisfied with the revision. No comments remained.

Reviewer #2 (Remarks to the Author):

The revised paper has addressed all comments and suggestions by me.

The paper now is clear and publishable.

The novelty has been reinforced during the revision.

The paper is indeed a pioneering one in applying SNN and neuromorphic devices in clinical medicine and I will suggest that the paper is highlighted for its novelty in this journal.